# The impact of extended reality cognitive behavioral therapy on mental disorders among children and youth: A systematic review and meta-analysis protocol

Madeline Li[1], Jamin Patel[1,2], Tarun Reddy Katapally [1,2,3]*

1 DEPtH Lab, School of Health Studies, Faculty of Health Sciences, Western University, London, Ontario, Canada, 2 Department of Epidemiology and Biostatistics, Schulich School of Medicine and Dentistry, Western University, London, Ontario, Canada, 3 Children's Health Research Institute, Lawson Health Research Institute, London, Ontario, Canada

* tarun.katapally@uwo.ca

## Abstract

### Background

The prevalence of mental disorders among children and youth has significantly increased, with rising rates of anxiety, depression, and other psychological disorders globally. Despite the widespread adoption of cognitive behavioral therapy (CBT) as a standardized treatment for various mental disorders, its efficacy can be constrained due to limited patient engagement, lack of commitment, and stigma, all challenges pronounced among children and youth. In this context, extended reality (XR) technologies (including virtual, augmented, and mixed reality) have emerged as innovative therapeutic tools offering immersive and engaging environments to overcome the limitations of traditional CBT.

### Objectives

This protocol aims to outline the methodology for conducting a systematic review and meta-analysis to evaluate the impact of XR-CBT on symptoms of mental disorders among children and youth.

### Methods

This systematic review and meta-analysis will follow PRISMA-P 2015 guidelines. A comprehensive search will be conducted in PsycINFO, PubMed, EMBASE, Scopus, and Web of Science to identify relevant studies published between January 2014 and June 2024. Eligible studies must involve children and youth (ages 24 years or younger) diagnosed with a mental disorder (e.g., anxiety, depression, ADHD, PTSD) and compare XR-CBT interventions (virtual, augmented, or mixed reality) with traditional therapy or control groups (e.g., no treatment). The primary outcome will be the change in symptoms of mental disorders, measured using standardized instruments (e.g., PHQ-9, GAD-7, PSS). Data will be extracted on post-intervention means, standard deviations, and 95% confidence intervals.

**Data availability statement:** No datasets were generated or analysed during the current study. All relevant data from this study will be made available upon study completion.

**Funding:** This research was funded by the Canada Research Chairs program which funds Dr. Tarun Katapally's research program. The funders had no role in study design, data collection and analysis, decision to publish, or preparation of the manuscript.

**Competing interests:** The authors have declared that no competing interests exist.

Effect sizes, calculated using Hedges' g, will be pooled with a random-effects model. Moreover, an a priori meta-regression within a random-effects framework will be conducted to examine how study-level characteristics influence effect sizes and address heterogeneity across studies. Heterogeneity will be assessed using the $I^2$ statistic and the Cochran's Q test. Risk of bias in individual studies will be evaluated using the Cochrane risk-of-bias tool.

## Conclusions

This protocol establishes a structured approach for assessing the efficacy of XR-CBT interventions on mental disorders among children and youth. The results of the systematic review and meta-analysis will fill a gap in current research and inform future therapeutic applications for mental health interventions among children and youth.

## 1. Introduction

Mental disorders constitute a significant public health challenge, particularly among children and youth (e.g., ages 24 years and younger [1–3]) whose prevalence is notably high [4–8]. Studies indicate that 1 in 7 adolescents experience a mental disorder in any given year [9], which profoundly affects their development [10], academic performance [11], and overall quality of life [12]. Moreover, mental disorders that originate in youth often extend into adulthood, as approximately 50% of all lifetime mental disorders have been shown to begin by the age of 14 years [13]. Furthermore, the global economic burden of mental disorders is substantial, amounting to $1 trillion annually [14]. This underscores the need for more innovative therapeutic interventions to address the unique mental health challenges faced by children and youth.

An established therapeutic approach for treating various mental disorders is cognitive behavioral therapy (CBT). CBT is based on the cognitive theory that negative thought patterns can lead to emotional distress and behavioral issues [15–17]. Through structured therapy sessions, CBT enables individuals to recognize these maladaptive thoughts, challenge their accuracy, and modify behaviors [18]. CBT encompasses different subtypes of therapeutic interventions, including exposure therapy [19], trauma-focused CBT [20], and behavioral activation therapy [21], which can be effective in treating anxiety disorders [22], phobias [23], depression [24], and post-traumatic stress disorder (PTSD) [23]. Despite being a gold-standard therapeutic intervention [25], CBT has several limitations. CBT requires active participation and a high level of commitment from patients [26], which can be challenging for children and youth. Furthermore, the traditional face-to-face format may not engage younger patients effectively, as for certain types of phobias or anxieties, it can be difficult to provide the necessary exposures in a controlled and practical manner during therapy sessions [27,28]. Additionally, seeking traditional face-to-face therapies such as CBT often carries a stigma, especially among children and youth [29], who may be reluctant to receive mental health care due to societal perceptions and judgement from peers and family [30–32]. These limitations highlight the need for more adaptable and engaging therapeutic approaches to effectively address mental disorders among children and youth.

A promising solution to these limitations is the use of extended reality CBT (XR-CBT). This innovative intervention employs advanced technologies, such as virtual reality (VR), augmented reality (AR), or mixed reality (MR) [33], to provide patients with a controlled and safe virtual environment to confront and manage their mental disorders [34]. XR-CBT serves as an umbrella term encompassing various specific approaches, with VR-CBT [35–37] and VR

exposure therapy (VRET) [38–40] being the most utilized methodologies. These XR-CBT interventions build on core CBT principles by creating environments that incorporate the immersive nature of XR to create realistic scenarios where patients can practice coping strategies, desensitize themselves to anxiety-provoking stimuli, and receive real-time feedback from their therapists [41–44]. The immersive nature of XR-CBT supports the cognitive-behavioral process of thought-behavior-emotion integration by directly engaging the patient in realistic scenarios where learned skills can be applied, which can strengthen therapeutic outcomes [35,37].

Moreover, previous evidence has shown XR-CBT interventions to be more acceptable and lead to higher satisfaction levels compared to traditional therapeutic approaches [45,46]. Among patients with specific phobias, the refusal rate for VRET was 3%, substantially lower than the 27% refusal rate observed with in-vivo exposure [46]. In a study examining soldier's attitudes on technology-based approaches in mental health care treatment, 19% percent of individuals who indicated they were unwilling to speak with a counsellor in person reported a willingness to use VR approaches for accessing mental health care, indicating that VR interventions may help overcome certain barriers to treatment [47]. By integrating these advanced technologies, XR-CBT has the potential to enhance the traditional CBT framework, making therapeutic interventions more engaging for children and youth [48].

Furthermore, the advancement of XR has been applied to treat various mental disorders among children and youth. For instance, VR therapy interventions have been used to address anxiety disorders among children and youth, particularly for specific phobias where traditional exposure therapy is difficult to conduct [49]. Current evidence suggests that VR interventions are an effective and well-accepted treatment approach for addressing psychological distress among adolescents [50]. Moreover, within hospital settings, VR has been used as an engaging therapeutic method to alleviate pain and anxiety among in-patient adolescents [51]. However, current evidence on the benefits of VR lacks consensus – some studies report benefits such as enhanced cognitive, motivational, emotional, and social development, while others highlight risks including addiction, anxiety, emotional distress, sleep issues, cyber sickness, and obesity [52]. Given the double-edged nature of XR interventions for mental disorders, the high prevalence of mental disorders among children and youth [9,12,13], and the limitations of traditional CBT [26,29–32], it is crucial to investigate the efficacy of XR-CBT for this demographic.

Currently, no systematic reviews have explored the impact of XR-CBT on mental disorders among children and youth. Prior reviews have examined the effectiveness of XR-CBT therapy interventions; however, they have primarily focused on general populations rather than children and youth [38,53,54]. While reviews on XR interventions among children and youth exist, they have focused on specific mental disorders, such as anxiety [49] or schizophrenia [55], rather than evaluating the broader applications of XR-CBT. Notably, reviews of XR interventions for mental disorders among children and youth have predominantly explored the VR dimension of XR [49–51,55–57], with limited focus on other XR interventions such as AR or MR. Furthermore, many systematic reviews have concentrated on other forms of XR therapies, such as XR distraction and relaxation therapy [58–60], rather than XR-CBT. Thus, this systematic review and meta-analysis will aim to quantitatively assess the impact of XR-CBT interventions on symptoms of mental disorders among children and youth. Moreover, this review will also generate recommendations for future development, implementation, and evaluation of XR-CBT that aims to treat child and youth mental disorders.

## 2.  Methods

This systematic review and meta-analysis protocol follows the Preferred Reporting Items for Systematic Review and Meta-Analysis Protocols (PRISMA-P) 2015 guidelines (S1 Fig) [61].

## 2.1. Research question

To what extent do XR-CBT interventions impact the symptoms of mental disorders in children and youth compared to traditional therapies and/or a control (e.g., no treatment)?

**2.1.1. Hypothesis.** XR-CBT shows a greater effect on symptoms associated with mental disorders in children and youth compared to traditional therapies and the control condition (e.g., no treatment) at post-intervention.

**2.1.2. Eligibility criteria.** This review will include studies involving children and youth ages 24 years and younger [1–3]. Participants must have a diagnosis of a mental disorder (e.g., anxiety, depression, attention-deficit/hyperactivity disorder (ADHD), PTSD, etc.) to be eligible for inclusion. This review broadly considers mental disorders as defined by the Diagnostic and Statistical Manual of Mental Disorders, Fifth Edition (DSM-5), which provides the standard classification of mental disorders used by mental health professionals [62]. The DSM-5 outlines specific criteria for diagnosing mental disorders, including disorders related to mood (such as depression), anxiety (such as generalized anxiety disorder, phobias, and social anxiety), and other conditions like ADHD, obsessive-compulsive disorder (OCD), and PTSD [62]. Each disorder is characterized by a unique set of symptoms and diagnostic criteria, and the review will consider the diversity of these conditions when examining the potential applications of XR-CBT across different mental disorders. There will be no restrictions on the severity or duration of the mental disorder. Mental disorder outcomes will be measured through different instruments, such as the Perceived Stress Scale (PSS) [63,64], The Patient Health Questionnaire-9 (PHQ-9) [65,66], and the Generalized Anxiety Disorder scale (GAD-7) [67,68]. The intervention that will be examined in this review is XR-CBT, which uses immersive technologies (VR, AR, MR) to deliver CBT. All studies must examine the effects of XR-CBT interventions on mental disorders. The alternatives against which the intervention (XR-CBT) will be compared in this review include stand-alone CBT, other traditional therapies, and/or a control group, such as no treatment or a placebo.

Moreover, the review will include experimental study designs such as randomized controlled trials (RCTs), quasi-randomized controlled trials (quasi-RCTs), and controlled clinical trials (CCTs) that examine the effects of XR-CBT interventions on mental disorders in children and youth. Studies that only explore participants' experiences or understanding of XR-CBT interventions and those that do not specifically assess the effects of XR-CBT on mental disorders (e.g., feasibility studies, developmental studies) will be excluded. Additionally, this review will only examine studies published in English. In consideration of ongoing technological advancements, this review will aim to present a comprehensive overview of the latest applications of XR-CBT by analyzing studies published within the past decade, specifically from January 2014 to June 2024. Table 1 presents a detailed overview of the inclusion and exclusion criteria of this systematic review.

## 2.2. Study records

**2.2.1. Data sources.** The review will use the following databases to analyze key studies on this topic: Embase, PubMed, Web of Science, PsycInfo, and Scopus. Embase and PubMed will be used as a source for biomedical research, providing valuable clinical research on mental disorders and medical devices. PsycInfo will be used to gain comprehensive coverage of psychology and behavioral science literature. As this review will include components of psychology, medicine, and technology, Scopus and Web of Science will be used to provide a broad and inclusive perspective from various fields. The advanced search tool will be utilized for all databases to ensure precise and comprehensive retrieval of relevant studies.

**Table 1. Inclusion and exclusion criteria for this systematic review.**

| Concept | Inclusion | Exclusion |
|---|---|---|
| Age | Ages 0-24 years | All other ages |
| Therapeutic intervention | Studies using XR-CBT to address mental disorders among children and youth, where XR technology is central to the intervention. (e.g., VR-CBT, VRET, AR-CBT, ARET, MR-CBT, XR-CBT, XRET) | Studies using non-immersive CBT, non-CBT therapies via XR, or those focusing only on feasibility or user experience without evaluating mental disorders. |
| Comparator(s) | Studies that will be included in the review must compare XR-CBT with traditional therapies, such as in-person therapy, or stand-alone CBT. Additionally, studies that include a control group, such as those receiving no treatment, placebo, or standard care, will also be eligible for inclusion. | Studies that do not compare XR-CBT interventions with an alternative therapy or control will be excluded. |
| Mental disorders | Changes in mental disorder symptoms | Physical disorders/other health issues not relating to mental disorders, economic, cognitive, and physical outcomes not related to mental disorders |
| Language | English | All other languages |
| Publication status | Peer-reviewed literature | Grey literature |
| Date of publication | January 2014 to June 2024 | All other dates |
| Type of study | Studies with experimental designs, including RCTs, quasi-RCTs, and CCTs, that assess the efficacy of XR-CBT interventions in treating mental disorders. | Observational studies, case studies, feasibility studies, developmental studies, or studies that do not utilize experimental or quasi-experimental designs. |

**2.2.2. Search strategy.** In consultation with domain-specific librarians to ensure credibility, three principal thematic categories were incorporated into the search strategy: therapeutic intervention, mental disorder, and age. A key concepts map is provided in Table 2. Moreover, the reference list of the peer-reviewed literature included will also be manually examined to identify any relevant articles that might have been missed by the initial search criteria. The full search strategy for all databases can be found in S2 Fig.

**2.2.3. Data screening and extraction.** After implementing the search strategy in each database, the results will be transferred into Covidence software for systematic screening and management. In the initial stage, duplicates will be removed, and the remaining studies will be assessed for the inclusion criteria based on their titles and abstracts by two independent reviewers, thus ensuring reliability. Any conflicts will be discussed and resolved between the two reviewers. In the next stage, the full texts of studies selected in the initial stage will be thoroughly reviewed by both independent reviewers. If any studies do not meet the inclusion criteria, they will be discussed between the reviewers before the data extraction process. Moreover, the Cochrane risk-of-bias tool will be used to assess the risk of bias for individual studies included in the systematic review and meta-analysis [69]. Specifically, the Cochrane risk-of-bias tool will evaluate potential biases in randomization, allocation concealment, blinding, incomplete outcome data, and selective outcome reporting [69]. Covidence will be utilized to present the results using the PRISMA 2020 four-phase flow diagram [70] and to systematically extract data from each of the included studies in the specified categories: year of publication, study design, aim of study, sample size of the study population, baseline characteristics of the study population, mental disorder, and type of XR-CBT intervention.

Furthermore, this study does not involve the collection of primary data and therefore does not require ethics approval. However, we will assess the ethical considerations of included studies using a framework adapted from Hawker's quality assessment and risk of bias tool [71], focusing on ethics (e.g., were ethical issues, including confidentiality, sensitivity, and consent, properly addressed, and was necessary approval obtained?), the researcher-participant relationship (e.g., was this relationship adequately considered?), and bias (e.g., was the researcher reflexive or aware of their own bias?). Hawker's framework was selected as it uniquely supports the evaluation of ethical considerations, including the appropriateness of

**Table 2. Key concepts map.**

| Thematic category | Search terms |
|---|---|
| Therapeutic intervention | Virtual Reality Cognitive Behavioral Therapy, Virtual Reality Cognitive Behavioral Therapy, Virtual Reality CBT, VRCBT, Virtual Reality Exposure Therapy, VRET, Virtual reality training, cognitive behavior* therapy with virtual reality, Augmented Reality with Cognitive Behavioral Therapy, Augmented Reality with Cognitive Behavioral Therapy, AR-CBT, ARET, Mixed Reality with Cognitive Behavioral Therapy, Mixed Reality with Cognitive Behavioral Therapy, MR-CBT, extended reality with cognitive behavioral therapy, extended reality with cognitive behavioral therapy, XR-CBT, XRET |
| Mental disorder | Mental Health, Psychological Wellbeing, Wellbeing, Anxiet*, Phobia, OCD, Depress*, PTSD, Disorder*, Behaviour*, Behavior, Psycholog* |
| Age | Youth, Adolescent*, Teen*, Student*, Child*, college student* |

the consent process, the protection of participant privacy, and the safeguarding against potential risks associated with XR-CBT interventions [52]. These aspects are particularly relevant for studies involving XR-CBT, given the emerging nature of these interventions [72,73] and the involvement of vulnerable populations such as youth and children [74–77].

**2.2.4. Data synthesis and analysis.** *Pre-specified analyses:* All data analyses and statistical modeling will be conducted using R 4.4.2 [78]. We will extract post-intervention means, standard deviations (SDs), and 95% confidence intervals (CIs) from each study, and we will calculate the effect sizes using Hedges' g. Hedges' g is an estimator of effect sizes that corrects for small sample bias [79], making it particularly appropriate for this study given the potential inclusion of smaller studies. Additionally, greater weight will be given to RCTs, as they are generally considered the gold standard in minimizing selection bias through randomization and controlled conditions [80,81]. In contrast, quasi-experimental studies can introduce potential biases, such as selection bias due to the lack of random assignment [82], and performance bias due to inconsistent intervention delivery across non-randomized study groups [83]. Moreover, a random-effects model will be used to pool the calculated effect sizes across studies. This model was selected to account for variations between studies and individuals, making it appropriate for addressing the heterogeneity expected across studies [84]. Unlike a fixed-effects model, which assumes a single common effect size, the random-effects model accommodates variations in effect sizes due to differences in study designs, populations, and interventions [84]. Additionally, we will conduct an a priori meta-regression within the random-effects framework to explore how study-level characteristics (e.g., sample size, study quality) may influence effect sizes and to address heterogeneity across studies. This analysis will focus on specific outcomes from the DSM-5 measures related to mental disorder symptoms, provided that at least 10 studies are included. Moreover, heterogeneity will be assessed using the $I^2$ statistic and the Cochran's Q test, with 25%, 50%, and 75% indicating low, moderate, and high levels. Publication bias will be assessed using funnel plots and statistically tested using the Egger's regression test. When available, intention-to-treat (ITT) data will be used for the primary analysis. Additionally, any missing data or incomplete reporting will be addressed by contacting the study authors for clarification.

*Exploratory analyses:* For exploratory analyses, subgroup analyses will be performed to examine differences by study design, intervention type, and participant characteristics. To address variability in follow-up times across studies, we will categorize follow-up periods into short-term (0-6 months), medium-term (6-12 months), and long-term (>12 months) intervals for the outcomes of interest from the DSM-5 measures. These time categories are based on current evidence from behavioral intervention studies, which indicate that different follow-up durations reflect distinct phases of intervention effectiveness

and sustainability [85,86]. This approach will allow us to assess both the immediate, short-term effects of XR-CBT interventions—capturing early symptom reduction and treatment adherence [87]—as well as the long-term impacts, which are crucial for understanding sustained efficacy [88]. Additionally, we will conduct a post-hoc exploratory meta-regression to assess the association between factors such as age, gender, time since diagnosis, and the severity of symptoms associated with mental disorders. These meta-regressions will help identify potential moderators of intervention effects, enhancing our understanding of the variability in study outcomes. We will generate both pooled and stratified forest plots for the pre-specified and exploratory analyses. Moreover, sensitivity analyses will be conducted by excluding studies with small sample sizes or high risk of bias as assessed by the Cochrane risk-of-bias tool [69]. A "small" sample size will be defined as studies with fewer than 30 participants per group, aligning with common practices in meta-analyses [89,90].

## 3. Strengths and limitations

The systematic review and meta-analysis will evaluate XR-CBT interventions across a broad spectrum of mental disorders among children and youth. By focusing on a wide range of conditions, we aim to provide a comprehensive assessment of XR-CBT's efficacy and applicability. This review will include all studies on XR-CBT on mental disorders among children and youth, regardless of their results, thus minimizing the impact of selective reporting and enhancing the overall accuracy of our findings.

However, a potential limitation is that participating in XR-CBT interventions requires access to advanced technologies, which may not be available to certain populations, particularly in low-income areas [91,92]. This digital divide could lead to an underrepresentation of these groups in the studies included for this review. Additionally, another limitation of this review is the potential language bias from including only English-language studies, which may exclude relevant research in other languages. Moreover, this study exclusively focuses on XR-CBT interventions and does not assess other XR therapies.

## 4. Conclusion

The systematic review and meta-analysis will overview the existing evidence on the impacts of XR-CBT interventions on mental disorders among children and youth. Additionally, findings from this review will not only contribute to the existing body of literature but will also offer evidence-based recommendations for policy-makers and mental health practitioners on how to effectively implement XR-CBT interventions. This includes considerations for integrating these technologies into existing mental health care frameworks to ensure accessibility, while addressing the unique needs of young patients. Given the significant increase of mental disorders among this demographic [4–8] and the current constraints of traditional CBT [26–32], there is an urgent need for innovative and more effective therapeutic approaches. By advancing our understanding of XR-CBT, this research has the potential to initiate a shift in our current delivery of mental health care for children and youth, working to sustain long-term positive mental health outcomes.

## Supporting information

**S1 Fig. PRISMA-P (Preferred Reporting Items for Systematic review and Meta-Analysis Protocols) 2015 checklist: recommended items to address in a systematic review protocol.** (PDF)

**S2 Fig. Search strategy to be used for all databases (PubMed, Embase, PsycINFO, Web of Science, and Scopus).**
(PDF)

## Acknowledgments

We would like to thank the University of Western Ontario librarians for their help in creating our literature search strategy.

## Author contributions

**Conceptualization:** Madeline Li, Tarun Reddy Katapally.

**Funding acquisition:** Tarun Reddy Katapally.

**Investigation:** Madeline Li, Jamin Patel, Tarun Reddy Katapally.

**Methodology:** Jamin Patel.

**Supervision:** Tarun Reddy Katapally.

**Visualization:** Madeline Li.

**Writing – original draft:** Madeline Li, Jamin Patel.

**Writing – review & editing:** Madeline Li, Jamin Patel, Tarun Reddy Katapally.

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
