## [Decision Letter · Decision Letter 0]

10 Sep 2024

PONE-D-24-35309The impact of extended reality cognitive behavioral therapy on mental health outcomes among children and youth: a systematic review protocolPLOS ONE

Dear Dr. Katapally,

Thank you for submitting your manuscript to PLOS ONE. After careful consideration, we feel that it has merit but does not fully meet PLOS ONE’s publication criteria as it currently stands. Therefore, we invite you to submit a revised version of the manuscript that addresses the points raised during the review process.

We look forward to receiving your revised manuscript.

Kind regards,

Mohammad Jamil Rababa

Academic Editor

PLOS ONE

Journal Requirements:

“Canada Research Chairs”

Reviewers' comments:

Reviewer's Responses to Questions

**Comments to the Author**

1. Does the manuscript provide a valid rationale for the proposed study, with clearly identified and justified research questions?

Reviewer #1: Partly

Reviewer #2: Yes

Reviewer #3: Yes

Reviewer #4: Partly

2. Is the protocol technically sound and planned in a manner that will lead to a meaningful outcome and allow testing the stated hypotheses?

Reviewer #1: Yes

Reviewer #2: Yes

Reviewer #3: Partly

Reviewer #4: No

3. Is the methodology feasible and described in sufficient detail to allow the work to be replicable?

Reviewer #1: Yes

Reviewer #2: Yes

Reviewer #3: Yes

Reviewer #4: No

4. Have the authors described where all data underlying the findings will be made available when the study is complete?

Reviewer #1: No

Reviewer #2: Yes

Reviewer #3: Yes

Reviewer #4: No

5. Is the manuscript presented in an intelligible fashion and written in standard English?

Reviewer #1: Yes

Reviewer #2: Yes

Reviewer #3: Yes

Reviewer #4: Yes

6. Review Comments to the Author

You may also provide optional suggestions and comments to authors that they might find helpful in planning their study.

Reviewer #1: Thank you for submitting your systematic review protocol titled "The Impact of Extended Reality Cognitive Behavioral Therapy on mental health outcomes among children and youth." This protocol addresses an important and timely topic, exploring the innovative use of extended reality (XR) technologies in cognitive behavioral therapy (CBT) for children and youth, a demographic experiencing significant mental health challenges.

Below are specific comments and suggestions to address before this protocol can be accepted for publication:

(1) Rationale and Research Questions: The manuscript provides a strong rationale for exploring XR-CBT and clearly outlines the research questions. However, further elaboration on how XR-CBT uniquely addresses the mental health needs of children and youth compared to traditional CBT would strengthen the justification for the study.

(2) Technical Soundness and Methodology: The methodology is robust and follows PRISMA-P guidelines, ensuring a structured approach to the systematic review. However, providing additional details on the data synthesis method (e.g., narrative synthesis or meta-analysis) would enhance the transparency of the planned analyses.

The current inclusion and exclusion criteria are clear and appropriate. Consider including a brief discussion on any exploratory analyses planned or how potential methodological challenges will be managed.

(3) Feasibility and Replicability: The protocol describes the methodology sufficiently to allow replication by other researchers. The use of Covidence software, independent reviewers, and a detailed search strategy are all strengths that support the feasibility of the review. No major changes are required in this regard.

(4) Data Availability: The protocol currently does not specify where the data underlying the findings will be made available upon completion of the review. Please include a statement detailing the data-sharing plan, such as depositing the data in a public repository, to comply with PLOS ONE’s data policy. This will ensure transparency and accessibility of the review's findings.

(5) Presentation and Language: The manuscript is well-written and presented, with minimal language errors. The writing is clear, coherent, and professional, making the manuscript easy to understand. Only minor language polishing is needed to improve readability and flow.

(6) Additional Suggestions: Consider adding a brief theoretical framework that connects XR technologies with CBT principles to strengthen the conceptual basis of the review.

Expanding on the future implications and potential challenges of XR-CBT (e.g., ethical considerations, cost, and accessibility issues) would provide a more comprehensive outlook on its practical application.

Overall, the protocol is well-constructed and addresses a critical gap in the literature. Minor revisions, as outlined above, will enhance the quality and transparency of the manuscript, making it suitable for publication. We look forward to seeing the final version of this important work.

Thank you for your contributions to this emerging area of mental health research.

Reviewer #2: The paper titled "The impact of extended reality cognitive behavioral therapy on mental health outcomes among children and youth: a systematic review protocol" addresses the growing concern of mental health issues in children and youth, particularly the increasing rates of anxiety, depression, and other psychological disorders. The authors highlight the limitations of traditional cognitive behavioral therapy (CBT), which often struggles with patient engagement and stigma, especially among younger populations. In response to these challenges, they propose investigating extended reality (XR) technologies—encompassing virtual reality, augmented reality, and mixed reality—as innovative therapeutic tools that may enhance the effectiveness of CBT by providing immersive and engaging environments.

The systematic review aims to evaluate the efficacy of XR-CBT interventions specifically for children and youth, an area that has not been systematically reviewed before. The authors outline a comprehensive literature search strategy utilizing five electronic databases to gather relevant studies. They will assess the impact of XR-CBT on mental health outcomes, explore the customization of these interventions, and consider their integration into digital mental health platforms. The findings from this review are expected to contribute valuable insights into the potential of XR-CBT to address the escalating mental health crisis among younger populations, ultimately informing future therapeutic practices and research in this emerging field.

In my opinion, the idea of this Research article is really interesting and the authors’ fascinating observations on this timely topic may be of interest to the readers of Plos One. However, some comments, as well as some crucial evidence that should be included to support the authors’ argumentation, need to be addressed to improve the quality of the article, its adequacy, and its readability prior to the publication in the present form.

Strengths:

• The paper addresses a critical and timely issue, given the rising prevalence of mental health disorders among children and youth. The integration of XR technologies into therapeutic practices is an innovative approach that could potentially enhance engagement and treatment efficacy.

• The objectives of the systematic review are clearly outlined, focusing on the efficacy, customization, integration into digital platforms, and future recommendations for XR-CBT. This provides a comprehensive framework for the review.

Please consider the following comments:

• The introduction could benefit from a more extensive literature review that discusses existing studies on XR-CBT, particularly those involving children and youth. This would help establish a stronger foundation for the need for this systematic review.

• While the paper mentions the databases to be used for the literature search, it lacks specific details about the search strategy (e.g., keywords, inclusion/exclusion criteria). I would ask the authors to provide this information to enhance the reproducibility of the review.

• The paper should address potential biases in the selection of studies, such as publication bias or language bias. Including a plan for assessing the risk of bias in the included studies would strengthen the review's credibility.

• Although the paper mentions the need for an ethics statement, it does not elaborate on how ethical considerations will be handled, particularly given the vulnerable population involved (children and youth).

• The authors state that no datasets will be generated or analyzed during the current study. However, they should clarify how they plan to share any findings or insights from the review, as transparency in data availability is increasingly important in research.

• The review should specify how the impact of XR-CBT will be measured. Authors should better define clear metrics or outcomes for evaluating mental health outcomes will enhance the study's applicability and relevance.

Reviewer #3: Recommendations for Manuscript ID PONE-D-24-35309 Title: “ The impact of extended reality cognitive behavioral therapy on mental health outcomes among children and youth: a systematic review protocol” for the Plos One.

General Comments

From my point of view, it is a very interesting topic and simultaneously it seems that to the best of my knowledge is an empirical study aims to evaluate the impact of XR-CBT on mental health outcomes among children and youth. A comprehensive literature search was conducted using five electronic databases: PsycINFO, PubMed, EMBASE, Scopus, and Web of Science. The search strategy will include terms related to virtual reality, augmented reality, mixed reality, cognitive behavioral therapy, exposure therapy, youth, children, and mental health. Two independent reviewers will evaluate the search results from the five selected databases based on the pre-determined search strategy. Findings from this study will provide insights regarding the potential of XR-CBT to enhance mental health outcomes, which can contribute to addressing the escalating mental health crisis among children and youth.

The paper contains the following sections: Introduction, Methods, Limitations, Conclusion.

However, I find some recommendations:

1. The Manuscript needs careful English proofreading because there are some shortcomings. For instance, the article “the” is sometimes missing in front of nouns, the message in some paragraphs is not clear enough. It looks like the first part was written by one author with a greater command of the English language, and the rest of the paper was written by someone else. The numerous grammar errors made this a difficult paper to read. It was strange to see the authors refer to tables that were not submitted. I was unable to find any supplementary material to the submission, so I think this was truly omitted by the authors. Please read the manuscript carefully.

2. The abstract must contain the main purpose of the paper, the research method used in the research and the main contributions.

3. The last section must contains Conclusions and Policy Implications.

4. It would be very useful to add the "Introduction" section and the purpose, objectives and hypothesis of the research. I consider that a weak point of the paper is that the authors did not show the novelty of the paper compared to other works. That is why, I consider that the introduction should specify the novelty of the paper compared to other papers published in this area.

5. The research is well based on science and the results are in agreement with the theoretical part. The paper is well written and easy to read.

6. The Literature Review section cannot be missing from the paper.

7. The model applied to the analyzed data is correctly used in the analysis undertaken, it is a strength point of this paper.

8. The research is well based on science and the results are in agreement with the theoretical part. The model applied to the analyzed data is correctly used in the analysis undertaken, it is a strength point of this paper.

9. Authors must present the software used in data processing (STATA, Eviews, etc)

10. In the same time, the authors must present a minimum an empiric model.

11. I recommend the authors to refer to other recent works indexed in Web of Science, because the papers cited works is not enough for a scientific paper. In my opinion, the authors must cite other papers regarding this subject.

1.

12. The conclusions at the end of the paper should be expanded showing the economic policy implications of the research results.

In conclusion, the article should be improve. It should also be enhanced with a review of the literature adequate to the subject and a broader interpretation and commentary of the research results.

Reviewer #4: Dear colleagues,

This review’s topic is fascinating, and will surely be of interest to the wider scientific community. The quality of writing is of a high standard, and the introduction and study context is generally clear and well-supported. However, I recommend a major re-conceptualisation and rewrite regarding the specifics of the review. In particular, I am concerned how particular constructs are defined (or lack thereof), and what outcome measurements are of interest for analysis. I recommend the authors follow pre-registration guidance at Prospero and/or OSF to tightly define the review's scope before proceeding any further. Currently, the protocol is close to being broad enough to perform a scoping review; however as a systematic review and meta-analysis the study design needs major revisions before being accepted as a published report. I have provided comments within the text for areas of revision.

Kind regards

7. PLOS authors have the option to publish the peer review history of their article (what does this mean? ). If published, this will include your full peer review and any attached files.

**Do you want your identity to be public for this peer review?** For information about this choice, including consent withdrawal, please see our Privacy Policy .

Reviewer #1: No

Reviewer #2: No

Reviewer #3: No

Reviewer #4: **Yes: ** Andrew Danso

---

## [Author Response · Author response to Decision Letter 1]

3 Oct 2024

Reviewer #1

The manuscript provides a strong rationale for exploring XR-CBT and clearly outlines the research questions. However, further elaboration on how XR-CBT uniquely addresses the mental health needs of children and youth compared to traditional CBT would strengthen the justification for the study.

Thank you for your suggestion. We have now added more content to describe the limitations of traditional CBT and how XR-CBT offers benefits for addressing them, with specific findings from previous literature: “Moreover, previous evidence has shown XR-CBT interventions to be more acceptable and lead to higher satisfaction levels compared to traditional therapeutic approaches (42,43). Among patients with specific phobias, the refusal rate for VRET was 3%, substantially lower than the 27% refusal rate observed with in-vivo exposure (43). In a study examining soldier attitudes on technology-based approaches in mental health care treatment, 19% percent of individuals who indicated they were unwilling to speak with a counsellor in person reported a willingness to use VR approaches for accessing mental health care, indicating that VR may help overcome certain barriers to treatment (44).” Lines 96-103

The methodology is robust and follows PRISMA-P guidelines, ensuring a structured approach to the systematic review. However, providing additional details on the data synthesis method (e.g., narrative synthesis or meta-analysis) would enhance the transparency of the planned analysis.

Thank you for your feedback. We have now added a section on the data synthesis methods (lines 213 to 220): “All data analysis and statistical modeling will be conducted using RStudio (63). We will extract post-intervention means, standard deviations (SDs), and 95% confidence intervals (CIs) from each study, and we will calculate standardized mean differences (SMDs) for the effect size. A random-effects model will be used to pool the calculated effect sizes across studies. Forest plots will be generated to visually represent the pooled effect sizes with corresponding confidence intervals. Heterogeneity will be assessed using the I² statistic and Cochran’s Q test. Subgroup analysis will be performed to examine differences by study design, intervention type, and participant characteristics.”.

Consider including a brief discussion on any exploratory analysis planned or how potential methodological challenges will be managed.

Thank you for your suggestion. We added more details regarding our approach to our planned meta-analysis above (lines 213 to 220). We have also added content on how we will address potential methodological challenges (lines 220 to 225): “Sensitivity analysis will be conducted by excluding studies with small sample sizes or high risk of bias. Publication bias will be assessed using funnel plots and statistically tested using Egger’s regression test. When available, intention-to-treat (ITT) data will be used for the primary analysis. Additionally, any missing data or incomplete reporting will be addressed by contacting the study authors for clarification.”.

The protocol currently does not specify where the data underlying the findings will be made available upon completion of the review. Please include a statement detailing the data-sharing plan, such as depositing the data in a public repository, to comply with PLOS ONE’s data policy. This will ensure transparency and accessibility of the review's findings.

A statement on data availability has been included in our protocol (Lines 253 to 256): Post-intervention data used in this review, including standard deviations, means, 95% confidence intervals, and effect sizes, will be made publicly accessible upon publication. These data will be deposited in a publicly available repository to ensure transparency and replicability of the findings. However, we will not be collecting any primary data as a part of this review.

Only minor language polishing is needed to improve readability and flow.

Thank you for your feedback. We have proofread the manuscript to improve readability and flow.

Consider adding a brief theoretical framework that connects XR technologies with CBT principles to strengthen the conceptual basis of the review.

Thank you for your feedback, we understand the importance of connecting XR-CBT interventions with core CBT principles to strengthen the conceptual basis of our systematic review.

We have now defined the core principles of CBT: “CBT is based on the cognitive theory that negative thought patterns lead to emotional distress and behavioral issues (14-16). Through structured therapy sessions, CBT enables individuals to recognize these maladaptive thoughts, challenge their accuracy, and modify behaviors (17)”. (Lines 67 to 70)

Thereafter, we described how XR technologies can build on these theoretical principles (Lines 90 to 96): “These XR interventions builds on core CBT principles by creating immersive environments that incorporate the immersive nature of XR to create realistic scenarios where patients can practice coping strategies, desensitize themselves to anxiety-provoking stimuli, and receive real-time feedback from their therapists (38-41). The immersive nature of XR-CBT supports the cognitive-behavioral process of thought-behavior-emotion integration by directly engaging the patient in realistic scenarios where learned skills can be applied, which can strengthen therapeutic outcomes (26,34).”

Expanding on the future implications and potential challenges of XR-CBT (e.g., ethical considerations, cost, and accessibility issues) would provide a more comprehensive outlook on its practical application.

Thank you for your feedback. We have added the potential challenges of XR-CBT in the Introduction (Lines 112 to 118): “However, current evidence on the benefits of VR lacks consensus – some studies report benefits such as enhanced cognitive, motivational, emotional, and social development, while others highlight risks including addiction, anxiety, emotional distress, sleep problems, cyber sickness, and obesity (51). Given the double-edged nature of XR interventions for mental health, the high prevalence of mental disorders among children and youth (9,10,12), and the limitations of traditional CBT (25–30), it is crucial to investigate the efficacy of XR-CBT for this demographic.”.

Reviewer #2:

The introduction could benefit from a more extensive literature review that discusses existing studies on XR-CBT, particularly those involving children and youth. This would help establish a stronger foundation for the need for this systematic review.

Thank you for your feedback and the opportunity to expand on the background information regarding the use of XR therapies among children and youth. Our revised introduction now provides a more comprehensive literature review, presenting evidence of how XR therapies are utilized across various mental health issues in this population. For instance, we have now added (Lines 106 to 112): “The advancement of XR in mental health care has been applied to treat various mental disorders among children and youth. For instance, VR therapy interventions have been used to address anxiety disorders among children and youth, particularly for specific phobias where traditional exposure therapy is difficult to conduct (48). Moreover, current evidence suggests that VR interventions are an effective and well-accepted treatment approach for addressing psychological distress among adolescents (49) . For instance, within hospital settings, VR has been used as an engaging therapeutic method to alleviate pain and anxiety among in-patient adolescents (50).”.

Additionally, we have cited and described evidence underscoring the potential safety and ethical concerns associated with these interventions (lines 112 to 118). This balanced perspective provides a stronger foundation for our review, demonstrating the critical need for further exploration into both the benefits and challenges of XR therapies for children and youth.

Overall, we have added 10 additional references citing existing studies on XR-CBT to enhance our literature review.

While the paper mentions the databases to be used for the literature search, it lacks specific details about the search strategy (e.g., keywords, inclusion/exclusion criteria). I would ask the authors to provide this information to enhance the reproducibility of the review.

Thank you for your suggestion to include specific details about the search strategy, such as keywords and inclusion/exclusion criteria. Table 1 (Line 173) outlines the inclusion and exclusion criteria, and Table 2 (Line 191), details the key concepts and search terms used in our literature search. Nevertheless, we have now uploaded our full search strategy as a supplementary file.

The paper should address potential biases in the selection of studies, such as publication bias or language bias. Including a plan for assessing the risk of bias in the included studies would strengthen the review's credibility.

Thank you for your feedback. We would like to clarify that language bias has been addressed in the limitations section of our protocol, highlighting that studies included in our systematic review are limited to manuscripts published in English (lines 236 to 238). Moreover, we have included a statement that indicates how the Cochrane risk-of-bias tool will be used to assess the risk of bias for individual studies included in the systematic review (lines 201 to 204).

We have also added to the Strengths and Limitations: “This review will include all studies on XR-CBT on mental health outcomes among youth and children, regardless of their results.”(Lines 243 to 246). Moreover, we have now added to the Methods: “publication bias will be assessed using funnel plots and statistically tested using Egger’s regression test.” (lines 221 to 222).

Although the paper mentions the need for an ethics statement, it does not elaborate on how ethical considerations will be handled, particularly given the vulnerable population involved (children and youth).

Thank you for highlighting the importance of ethical considerations, particularly when discussing vulnerable populations such as children and youth. We have now added an ethics statement to our data screening and extraction section: “This study does not involve the collection of primary data and therefore does not require ethics approval. However, we will assess and report on the ethical considerations related to the studies included, such as informed consent and confidentiality, where applicable.” (Lines 209 to 211)

The authors state that no datasets will be generated or analyzed during the current study. However, they should clarify how they plan to share any findings or insights from the review, as transparency in data availability is increasingly important in research.

A statement on data availability has been included in our protocol. Specifically, we will ensure that post-intervention means, standard deviations, 95% confidence intervals, and standardized mean differences for the effect size of each study will be made publicly available in our manuscript. However, we will not be collecting any primary data as a part of this review. (Lines 253 to 256)

The review should specify how the impact of XR-CBT will be measured. Authors should better define clear metrics or outcomes for evaluating mental health outcomes will enhance the study's applicability and relevance.

Thank you for your valuable feedback. We will perform a meta-analysis to quantitatively synthesize the results from the included studies. We have now added a section on the data synthesis methods (lines 213 to 220): “All data analysis and statistical modeling will be conducted using RStudio (63). We will extract post-intervention means, standard deviations (SDs), and 95% confidence intervals (CIs) from each study, and we will calculate standardized mean differences (SMDs) for the effect size. A random-effects model will be used to pool the calculated effect sizes across studies. Forest plots will be generated to visually represent the pooled effect sizes with corresponding confidence intervals. Heterogeneity will be assessed using the I² statistic and Cochran’s Q test, with 25%, 50%, and 75% indicating low, moderate, and high levels. Subgroup analyses will be performed to examine differences by study design, intervention type, and participant characteristics.”

Moreover, we have now added a new section in the Methods to describe the mental health outcomes assessed in this study: “This review broadly considers mental disorders as defined by the Diagnostic and Statistical Manual of Mental Disorders, Fifth Edition (DSM-5), which provides the standard classification of mental disorders used by mental health professionals (55). The DSM-5 outlines specific criteria for diagnosing mental health conditions, including disorders related to mood (such as depression), anxiety (such as generalized anxiety disorder, phobias, and social anxiety), and other conditions like ADHD, obsessive-compulsive disorder (OCD), and post-traumatic stress disorder (PTSD). Each disorder is characterized by a unique set of symptoms and diagnostic criteria, and the review takes into account the diversity of these conditions when examining the potential applications of XR-CBT across different mental health issues.” (Lines 140 to 148)

Reviewer #3

The Manuscript needs careful English proofreading because there are some shortcomings. For instance, the article “the” is sometimes missing in front of nouns, the message in some paragraphs is not clear enough. It looks like the first part was written by one author with a greater command of the English language, and the rest of the paper was written by someone else. The numerous grammar errors made this a difficult paper to read. It was strange to see the authors refer to tables that were not submitted. I was unable to find any supplementary material to the submission, so I think this was truly omitted by the authors. Please read the manuscript carefully.

Thank you for your feedback and for highlighting the areas where the manuscript could be improved. We have carefully proofread the document prior to submission, but we will conduct another thorough review to ensure clarity and correct any grammatical issues, including the use of articles and any other shortcomings. We have also ensured that all necessary supplementary files are correctly included.

The abstract must contain the main purpose of the paper, the research method used in the research and the main contributions.

Thank you for your feedback and the opportunity to improve the clarity in our abstract.

We added the purpose of the paper on to lines 36 to 38: “Objectives: This systematic review and meta-analysis protocol aims to outline the methodology for conducting a systematic review and meta-analysis to evaluate the impact of XR-CBT on reducing symptoms of mental disorders among children and youth.”

We added details regarding the research methods to lines 39 to 50: “Methods: A comprehensive literature search will be conducted using five databases: PsycINFO, PubMed, EMBASE, Scopus, and Web of Science. The search strategy will include terms related to virtual reality, augmented reality, mixed reality, cognitive behavioral therapy, exposure therapy, youth, children, and mental health. Two independent reviewers will evaluate the search results from the five selected databases based on the pre-determined search strategy. Moreover, a meta-analysis will be conducted to assess the efficacy of XR-CBT in reducing symptoms of mental disorders among children and youth. Post-intervention standard deviations and means will be extracted, and 95% confidence intervals will be calculated for standardized mean differences. Heterogeneity will be assessed using the I² statistic and Cochran’s Q test.”

Finally, we added the main contributions to lines 51 to 54: “Conclusions: This protocol establishes a clear methodology for systematically assessing the impact of XR-CBT on mental disorders among children and youth, addressing a gap in existing research and contributing to solutions for the

---

## [Decision Letter · Decision Letter 1]

9 Oct 2024

PONE-D-24-35309R1The impact of extended reality cognitive behavioral therapy on mental disorders among children and youth: a systematic review and meta-analysis protocolPLOS ONE

Dear Dr. Katapally,

Thank you for submitting your manuscript to PLOS ONE. After careful consideration, we feel that it has merit but does not fully meet PLOS ONE’s publication criteria as it currently stands. Therefore, we invite you to submit a revised version of the manuscript that addresses the points raised during the review process.

We look forward to receiving your revised manuscript.

Kind regards,

Mohammad Jamil Rababa

Academic Editor

PLOS ONE

Journal Requirements:

Reviewers' comments:

Reviewer's Responses to Questions

**Comments to the Author**

1. Does the manuscript provide a valid rationale for the proposed study, with clearly identified and justified research questions?

Reviewer #1: Yes

Reviewer #2: Yes

Reviewer #3: Yes

Reviewer #4: Yes

2. Is the protocol technically sound and planned in a manner that will lead to a meaningful outcome and allow testing the stated hypotheses?

Reviewer #1: Yes

Reviewer #2: Yes

Reviewer #3: Yes

Reviewer #4: Partly

3. Is the methodology feasible and described in sufficient detail to allow the work to be replicable?

Reviewer #1: Yes

Reviewer #2: Yes

Reviewer #3: Yes

Reviewer #4: Yes

4. Have the authors described where all data underlying the findings will be made available when the study is complete?

Reviewer #1: Yes

Reviewer #2: Yes

Reviewer #3: Yes

Reviewer #4: Yes

5. Is the manuscript presented in an intelligible fashion and written in standard English?

Reviewer #1: No

Reviewer #2: Yes

Reviewer #3: Yes

Reviewer #4: Yes

6. Review Comments to the Author

You may also provide optional suggestions and comments to authors that they might find helpful in planning their study.

Reviewer #1: (1) The manuscript requires further proofreading. Minor grammatical and typographical errors affect readability. These need to be corrected to ensure the manuscript is clear and correct and meets PLOS ONE's standards for intelligibility. Ensure smooth transitions between sections and improve the overall flow.

(2) Clearly distinguish any exploratory aspects of the analysis. While the protocol outlines the methodology well, it would benefit from explicitly marking which parts of the analysis are exploratory to prevent potential flexibility in interpretation after the study is conducted.

(3) Double-check that all supplementary materials are fully included and correctly uploaded. This includes ensuring the full search strategy and supplementary data files are accessible and meet the journal’s requirements.

So, please Conduct a thorough language revision, clarify exploratory analysis aspects, and ensure proper inclusion and access to all supplementary files.

Reviewer #2: Dear Authors,

Thank you for the thorough revisions and for addressing the key points raised in the previous review. The manuscript has improved significantly, particularly with the expanded literature review and detailed methodology. I appreciate the inclusion of additional references and the clarification of the search strategy, which strengthens the overall rigor of the paper.

However, there are a few additional minor points that I would like to bring to your attention for further refinement of the manuscript:

• The addition to the introduction on XR therapies (lines 106-112) is strong. However, the authors might further highlight how this systematic review addresses gaps in previous research (perhaps adding a sentence or two) to emphasize its unique contribution.

• In the section discussing the use of a random-effects model for the meta-analysis (lines 213 to 220), it could be helpful to briefly explain why a random-effects model is chosen over a fixed-effects model.

• While the ethics statement mentions that the study does not involve primary data collection and will assess ethical considerations of included studies (lines 209-211), authors might want to elaborate slightly on how this assessment will be done—whether specific frameworks will be used or if it will be handled case by case.

• While the manuscript mentions subgroup analyses to explore potential sources of heterogeneity (lines 219-220), consider indicating whether any visual representations of these subgroup analyses (e.g., stratified forest plots) will be included in the results.

• Authors mentioned that sensitivity analyses will exclude studies with small sample sizes (lines 220-221), but it could help to briefly define what constitutes a "small" sample size in this context, or explain how you will determine the threshold.

• While authors do mention subgroup analyses by study design (lines 219-220), it might be helpful to expand slightly on why different designs (e.g., RCTs vs. quasi-experimental) could affect the results or introduce biases, just to emphasize the rationale for this analysis.

I believe that addressing these minor points will further enhance the clarity and rigor of your manuscript. Thank you once again for your hard work and thoughtful revisions. I look forward to seeing the final version of this important contribution to the field.

Best regards,

Reviewer

Reviewer #3: Accept the paper for publication. I carefully examined the changes made to the paper and found that the authors took into account the reviewer's observations.

Reviewer #4: Dear colleagues, this is an excellent revision which has addressed many previous weaknesses. As this has substantially improved I will provide recommendations for a more detailed analysis plan:

Can you explicitly state what your reported measures of effect will be (e.g., Cohen's d, Hedges' g etc) as to your primary research question?

It is important to specify timepoints for the outcomes of interest from the DSM-5 outcome measures. For example, are you interested in measurements taken immediately, to 10-minutes post-intervention? And why? Please explicitly state this.

As part of your synthesis plan, I recommend including a plan to perform an apriori meta-regression analysis on specific outcomes from the DSM-5 outcome measures relevant to mental health symptoms, especially if the number of studies is at minimum 10, included in the meta-analysis. I would also recommend you do a post-hoc exploratory meta-regression analysis on factors such as age, gender, and time since diagnosis (as these factors may be associated with symptoms associated with mental disorders).

The hypothesis needs a minor revision to, "XR-CBT shows a greater effect on symptoms associated with mental disorders in children and youth compared to traditional therapies and the control condition (e.g., no treatment) at post-intervention".

7. PLOS authors have the option to publish the peer review history of their article (what does this mean? ). If published, this will include your full peer review and any attached files.

**Do you want your identity to be public for this peer review?** For information about this choice, including consent withdrawal, please see our Privacy Policy .

Reviewer #1: No

Reviewer #2: No

Reviewer #3: No

Reviewer #4: No

---

## [Author Response · Author response to Decision Letter 2]

23 Oct 2024

Reviewer #1:

The manuscript requires further proofreading. Minor grammatical and typographical errors affect readability. These need to be corrected to ensure the manuscript is clear and correct and meets PLOS ONE's standards for intelligibility. Ensure smooth transitions between sections and improve the overall flow.

Thank you for highlighting the need for further proofreading. We have carefully reviewed the manuscript for grammatical and typographical errors and ensured smoother transitions between sections to enhance the overall flow.

Clearly distinguish any exploratory aspects of the analysis. While the protocol outlines the methodology well, it would benefit from explicitly marking which parts of the analysis are exploratory to prevent potential flexibility in interpretation after the study is conducted.

Thank you for your comment. We have now clearly identified the exploratory aspects of our analysis in the revised manuscript. Specifically, the exploratory analyses will include: 1) Subgroup Analyses: subgroup analyses will be performed to examine differences by study design, intervention type, and participant characteristics; 2) Post-hoc exploratory meta-regression: In response to another reviewer, we will conduct a post-hoc exploratory meta-regression to assess the association between factors such as age, gender, time since diagnosis, and the severity of symptoms associated with mental disorders; 3) Sensitivity Analyses: We will conduct sensitivity analyses by excluding studies with small sample sizes or high risk of bias to assess the robustness of our primary findings. These exploratory analyses are now clearly distinguished with a subtitle “exploratory analyses” (line 243) from our other “pre-specified analyses” (line 221).

Double-check that all supplementary materials are fully included and correctly uploaded. This includes ensuring the full search strategy and supplementary data files are accessible and meet the journal’s requirements.

Thank you for your helpful feedback. We carefully reviewed and confirmed that all supplementary materials are fully included and correctly uploaded. This includes ensuring the full search strategy for each database and supplementary data files are accessible and meet the journal’s requirements. Supplementary files noted in the manuscript (Figure S1 (line 137), and Figure S2 (line 194)) are uploaded using the portal's instructions to ensure proper submission of the supplementary files.

So, please Conduct a thorough language revision, clarify exploratory analysis aspects, and ensure proper inclusion and access to all supplementary files.

Thank you for your valuable feedback. We have addressed your concerns as follows: 1) Language Revision: We have conducted a thorough revision of the manuscript to enhance clarity, improve readability, and ensure consistency throughout. We carefully reviewed the text to eliminate any ambiguities and improve the overall presentation of the study; 2) Exploratory Analyses: We have now explicitly clarified which parts of the analysis are exploratory. Specifically, the subgroup analyses, sensitivity analyses, and post-hoc exploratory meta-regression are designated as exploratory. These are clearly distinguished from the pre-specified analyses in the revised manuscript; 3) Supplementary Files: All supplementary files have been properly included and made accessible. We have ensured that they are correctly referenced in the manuscript and available for review to ensure transparency and full access to the data.

Reviewer #2

The addition to the introduction on XR therapies (lines 106-112) is strong. However, the authors might further highlight how this systematic review addresses gaps in previous research (perhaps adding a sentence or two) to emphasize its unique contribution.

Thank you for your feedback on the addition regarding XR therapies. In response to your suggestion, we have added two sentences to further highlight how this systematic review addresses gaps in previous research and emphasizes its unique contribution. Additionally, we have cited previous systematic reviews and meta-analyses to support this (lines 121-130):

“Currently, no systematic reviews have explored the impact of XR-CBT on mental disorders among children and youth. Prior reviews have examined the effectiveness of XR-CBT therapy interventions; however, they have primarily focused on general populations rather than children and youth (35,52,53). While reviews on XR interventions among children and youth exist, they have focused on specific mental disorders, such as anxiety (48) or schizophrenia (54), rather than evaluating the broader applications of XR-CBT. Notably, most XR reviews on mental disorders among children and youth have predominantly explored the VR dimension of XR (48–50,54–56), with limited focus on other XR interventions such as AR or MR. Furthermore, many systematic reviews have concentrated on other forms of XR therapies, such as XR distraction and relaxation therapy (57–59), rather than XR-CBT.”

In the section discussing the use of a random-effects model for the meta-analysis (lines 213 to 220), it could be helpful to briefly explain why a random-effects model is chosen over a fixed-effects model.

Thank you for your comment. We have selected a random-effects model for the meta-analysis because it accounts for potential variations both between studies and among individuals. This model is particularly appropriate in the context of our review, as it adjusts for heterogeneity across studies, recognizing that the true effect size may differ due to variations in study design, populations, and interventions. In contrast to a fixed-effects model, which assumes a common effect size across all studies, the random-effects model provides a more appropriate estimate by accounting for variability across studies. To support this rationale, we have also cited [Dettori et al., 2022] in the revised manuscript, providing evidence that this approach is widely accepted in meta-analyses. We have updated the manuscript to reflect this clarification. (lines 228-233):

“Moreover, a random-effects model will be used to pool the calculated effect sizes across studies. This model was selected to account for variations between studies and individuals, making it appropriate for addressing the heterogeneity expected across studies (74). Unlike a fixed-effects model, which assumes a single common effect size, the random-effects model accommodates variations in effect sizes due to differences in study designs, populations, and interventions (74)”

While the ethics statement mentions that the study does not involve primary data collection and will assess ethical considerations of included studies (lines 209-211), authors might want to elaborate slightly on how this assessment will be done—whether specific frameworks will be used or if it will be handled case by case.

Thank you for your suggestion. We will assess the ethical considerations of included studies using a framework adapted from Hawker’s quality assessment and risk of bias, focusing on Ethics (Were ethical issues, including confidentiality, sensitivity, and consent, properly addressed, and was necessary approval obtained?), Researcher-participant relationship (Was it adequately considered), Bias (Was the researcher reflexive or aware of their own bias?) (Lines 214-219)

While the manuscript mentions subgroup analyses to explore potential sources of heterogeneity (lines 219-220), consider indicating whether any visual representations of these subgroup analyses (e.g., stratified forest plots) will be included in the results.

Thank you for your feedback. Thank you for your suggestion. We will generate both pooled and stratified forest plots for the pre-specified and exploratory analyses (lines 257-258). Thus, we will visually represent the subgroup analyses and explore potential sources of heterogeneity.

Authors mentioned that sensitivity analyses will exclude studies with small sample sizes (lines 220-221), but it could help to briefly define what constitutes a "small" sample size in this context, or explain how you will determine the threshold.

Thank you for your suggestion. In response, we have clarified that a "small" sample size in the context of our sensitivity analyses will be defined as studies with fewer than 30 patients per group. This threshold aligns with common practices in meta-analyses. (Lines 258-261)

While authors do mention subgroup analyses by study design (lines 219-220), it might be helpful to expand slightly on why different designs (e.g., RCTs vs. quasi-experimental) could affect the results or introduce biases, just to emphasize the rationale for this analysis.

Thank you for this valuable suggestion. We acknowledge that different study designs, such as RCTs and quasi-experimental studies, may introduce varying levels of bias. RCTs are often regarded as the gold standard in clinical research due to their randomization process, which minimizes selection bias. Therefore, we will give more weight to RCTs in our analyses (lines 226-228). This approach is consistent with established guidelines that emphasize the higher methodological rigor of RCTs in comparison to quasi-experimental studies (Higgins et al., 2019: https://training.cochrane.org/handbook).

Reviewer #4

Can you explicitly state what your reported measures of effect will be (e.g., Cohen's d, Hedges' g etc) as to your primary research question?

Thank you for your comment. For our primary analysis, we will report Hedges' g as the measure of effect size. Hedges' g is chosen because it adjusts for potential bias in studies with small sample sizes, providing a more accurate estimate in these cases. While Cohen’s d is widely used and effective for larger sample sizes, Hedges' g is preferred here due to the possibility of including smaller studies in our meta-analysis. We believe this will enhance the precision of our effect size estimates. The manuscript has been updated to reflect this (lines 224-226).

It is important to specify timepoints for the outcomes of interest from the DSM-5 outcome measures. For example, are you interested in measurements taken immediately, to 10-minutes post-intervention? And why? Please explicitly state this.

Thank you for your comment. To address variability in follow-up times across studies, we will categorize follow-up periods into short-term (0-6 months), medium-term (6-12 months), and long-term (>12 months) intervals for the outcomes of interest from the DSM-5 measures (Lines 245-253):

“To address variability in follow-up times across studies, we will categorize follow-up periods into short-term (0-6 months), medium-term (6-12 months), and long-term (>12 months) intervals for the outcomes of interest from the DSM-5 measures. These time categories are based on current evidence from behavioral intervention studies, which indicate that different follow-up durations reflect distinct phases of intervention effectiveness and sustainability (75,76). This approach will allow us to assess both the immediate, short-term effects of XR-CBT interventions—capturing early symptom reduction and treatment adherence—as well as the long-term impacts, which are crucial for understanding sustained efficacy (77).”

As part of your synthesis plan, I recommend including a plan to perform an apriori meta-regression analysis on specific outcomes from the DSM-5 outcome measures relevant to mental health symptoms, especially if the number of studies is at minimum 10, included in the meta-analysis. I would also recommend you do a post-hoc exploratory meta-regression analysis on factors such as age, gender, and time since diagnosis (as these factors may be associated with symptoms associated with mental disorders).

Thank you for your valuable suggestion. We will include an a priori plan to perform a meta-regression analysis on DSM-5 outcomes relevant to mental health symptoms if at least 10 studies are included in the meta-analysis. This will allow us to explore potential associations between study characteristics and effect sizes. Additionally, we will conduct a post-hoc exploratory meta-regression to assess how factors such as age, gender, and time since diagnosis may influence mental health symptoms. These meta-regressions will help identify possible moderators of the intervention effects, further enhancing our synthesis of the evidence. The manuscript has been updated to reflect this plan (lines 233-237, lines 253-257):

“Additionally, we will conduct an a priori meta-regression within the random-effects framework to explore how study-level characteristics (e.g., sample size, study quality) may influence effect sizes and address heterogeneity across studies. This analysis will focus on specific outcomes from DSM-5 measures related to mental disorder symptoms, provided that at least 10 studies are included.”

“Additionally, we will conduct a post-hoc exploratory meta-regression to assess the association between factors such as age, gender, and time since diagnosis and the severity of symptoms associated with mental disorders. These meta-regressions will help identify potential moderators of intervention effects, enhancing our understanding of the variability in study outcomes.”

The hypothesis needs a minor revision to, "XR-CBT shows a greater effect on symptoms associated with mental disorders in children and youth compared to traditional therapies and the control condition (e.g., no treatment) at post-intervention".

Thank you for your feedback. The original hypothesis was, "XR-CBT shows a greater effect on mental disorders compared to traditional therapies and the control condition (e.g., no treatment) at post-intervention." We appreciate your suggestion to specify the focus on symptoms associated with mental disorders in children and youth. We recognize that your suggested revision adds clarity and aligns well with the study’s scope. Therefore, we have revised the hypothesis to: "XR-CBT shows a greater effect on symptoms associated with mental disorders in children and youth compared to traditional therapies and the control condition (e.g., no treatment) at post-intervention." (lines 141-144)

---

## [Decision Letter · Decision Letter 2]

11 Nov 2024

PONE-D-24-35309R2The impact of extended reality cognitive behavioral therapy on mental disorders among children and youth: a systematic review and meta-analysis protocolPLOS ONE

Dear Dr. Katapally,

Thank you for submitting your manuscript to PLOS ONE. After careful consideration, we feel that it has merit but does not fully meet PLOS ONE’s publication criteria as it currently stands. Therefore, we invite you to submit a revised version of the manuscript that addresses the points raised during the review process.

We look forward to receiving your revised manuscript.

Kind regards,

Mohammad Jamil Rababa

Academic Editor

PLOS ONE

**Journal Requirements:**

Reviewers' comments:

Reviewer's Responses to Questions

**Comments to the Author**

1. Does the manuscript provide a valid rationale for the proposed study, with clearly identified and justified research questions?

Reviewer #1: Yes

Reviewer #2: Yes

2. Is the protocol technically sound and planned in a manner that will lead to a meaningful outcome and allow testing the stated hypotheses?

Reviewer #1: Yes

Reviewer #2: Yes

3. Is the methodology feasible and described in sufficient detail to allow the work to be replicable?

Reviewer #1: Yes

Reviewer #2: Yes

4. Have the authors described where all data underlying the findings will be made available when the study is complete?

Reviewer #1: Yes

Reviewer #2: Yes

5. Is the manuscript presented in an intelligible fashion and written in standard English?

Reviewer #1: Yes

Reviewer #2: Yes

6. Review Comments to the Author

You may also provide optional suggestions and comments to authors that they might find helpful in planning their study.

**Reviewer #1:**  Thank you for the comprehensive revisions, which have made the protocol clear and robust, effectively addressing prior reviewer comments. You have provided an excellent justification for the research question, highlighting the significance of this study in filling an important gap in digital mental health.

The methodology is well-detailed and thoughtfully planned, with clear inclusion criteria, controls, and justified model choices. The manuscript is well-written, and the data availability plan aligns with journal standards.

No further revisions are necessary, and the protocol is ready for acceptance.

**Reviewer #2:**  Dear Authors,

I want to express my gratitude for the detailed and thoughtful responses provided to the comments I raised in the second review of this manuscript.

Authors have addressed my concerns and queries in a clear and comprehensive manner, and the revised manuscript aligns more closely with the scientific rigor expected for publication in PLOS ONE.

Having reviewed the revised manuscript, I am largely satisfied with the changes that have been implemented. I have just a couple of additional comments that I believe could further refine the manuscript:

- While you provided valuable information on the adapted framework based on Hawker’s quality assessment, it may be beneficial to briefly explain why this framework is especially suitable for assessing studies on XR-CBT in children and youth, enhancing clarity for readers.

- The addition explaining the rationale for analyzing different study designs is helpful; still, I believe that a brief note on specific biases that quasi-experimental studies might introduce (such as selection or performance bias) could further reinforce your rationale for prioritizing RCTs.

With these minor additions, I believe the manuscript will be ready for publication.

Thank you again for your diligent work on this revision.

Best regards,

Reviewer

7. PLOS authors have the option to publish the peer review history of their article (what does this mean? ). If published, this will include your full peer review and any attached files.

**Do you want your identity to be public for this peer review?** For information about this choice, including consent withdrawal, please see our Privacy Policy .

Reviewer #1: No

Reviewer #2: No

---

## [Author Response · Author response to Decision Letter 3]

13 Nov 2024

Editor: Mohammad Jamil Rababa

1. Please review your reference list to ensure that it is complete and correct. If you have cited

papers that have been retracted, please include the rationale for doing so in the manuscript

text, or remove these references and replace them with relevant current references. Any

changes to the reference list should be mentioned in the rebuttal letter that accompanies your

revised manuscript. If you need to cite a retracted article, indicate the article’s retracted status

in the References list and also include a citation and full reference for the retraction notice.

Thank you for your comment. We have carefully reviewed and confirmed that all references

in our manuscript are complete and correct. Three references were updated to strengthen the

alignment with our manuscript's content. Additionally, we verified that no retracted articles

are included in our reference list.

Reviewer #2: Dear Authors,

1. While you provided valuable information on the adapted framework based on Hawker’s

quality assessment, it may be beneficial to briefly explain why this framework is especially

suitable for assessing studies on XR-CBT in children and youth, enhancing clarity for readers.

Thank you for your comment and the opportunity to provide clarity in our data extraction

process. The Cochrane risk-of-bias tool is our primary quality appraisal tool, and it will be

employed to evaluate bias risk for each study included in this systematic review and metaanalysis. The Cochrane risk-of-bias tool will examine potential biases related to randomization,

allocation concealment, blinding, incomplete outcome data, and selective reporting of

outcomes (lines 204-207).

In response to comments/suggestions of another reviewer to elaborate on our ethics

assessment, we used questions derived from the Hawker’s assessment to guide our ethics

evaluation, which Hawker’s does more comprehensively than Cochrane risk-of-bias tool (lines

212-218). We created our questions based on the following specific questions from Hawker’s

quality appraisal: Were ethical aspects, including required approvals, confidentiality, sensitivity,

and informed consent, comprehensively addressed? Additionally, was the interaction between

researchers and participants thoughtfully considered?

Hawker’s framework uniquely supports evaluating ethical considerations such as the

appropriateness of the consent process, the protection of participants' privacy, and the

safeguarding of participants from any potential risks associated with XR technology—factors

that are particularly pertinent for studies on emerging interventions like XR-CBT among

young participants. Hawker’s was not used as the main quality appraisal framework, but rather

one of the components was used to supplement and enhance our primary quality appraisal.

We have now added this revision in the manuscript to the data screening and extraction section

(lines 218-223): “Hawker’s framework was selected as it uniquely supports the evaluation of

ethical considerations, including the appropriateness of the consent process, the protection of

participant privacy, and the safeguarding against potential risks associated with XR-CBT

interventions (52). These aspects are particularly relevant for studies involving XR-CBT, given

the emerging nature of these interventions (71,72) and the involvement of vulnerable

populations such as youth and children (73–76).”

2. The addition explaining the rationale for analyzing different study designs is helpful; still, I

believe that a brief note on specific biases that quasi-experimental studies might introduce

(such as selection or performance bias) could further reinforce your rationale for prioritizing

RCTs.

Thank you for your comment and for highlighting the importance of addressing specific biases

associated with the different study designs in our meta-analysis. To strengthen our rationale

for prioritizing randomized controlled trials (RCTs) in this review, we have added a note on

potential biases introduced by quasi-experimental studies, including selection bias, which may

arise due to the lack of random assignment, and performance bias, which can result from

inconsistent intervention delivery across non-randomized groups. By prioritizing RCTs, we

aim to minimize these biases and ensure more robust conclusions on the efficacy of XR-CBT

interventions.

We have now added these revisions to the manuscript (lines 232-235): “In contrast, quasiexperimental studies can introduce potential biases, such as selection bias due to the lack of

random assignment (81), and performance bias due to inconsistent intervention delivery across

non-randomized study groups (82).”

---

## [Decision Letter · Decision Letter 3]

25 Nov 2024

The impact of extended reality cognitive behavioral therapy on mental disorders among children and youth: a systematic review and meta-analysis protocol

PONE-D-24-35309R3

Dear Dr. Katapally,

We’re pleased to inform you that your manuscript has been judged scientifically suitable for publication and will be formally accepted for publication once it meets all outstanding technical requirements.

Kind regards,

Mohammad Jamil Rababa

Academic Editor

PLOS ONE

Additional Editor Comments (optional):

Reviewers' comments:

Reviewer's Responses to Questions

**Comments to the Author**

1. Does the manuscript provide a valid rationale for the proposed study, with clearly identified and justified research questions?

Reviewer #2: Yes

2. Is the protocol technically sound and planned in a manner that will lead to a meaningful outcome and allow testing the stated hypotheses?

Reviewer #2: Yes

3. Is the methodology feasible and described in sufficient detail to allow the work to be replicable?

Reviewer #2: Yes

4. Have the authors described where all data underlying the findings will be made available when the study is complete?

Reviewer #2: Yes

5. Is the manuscript presented in an intelligible fashion and written in standard English?

Reviewer #2: Yes

6. Review Comments to the Author

You may also provide optional suggestions and comments to authors that they might find helpful in planning their study.

Reviewer #2: Dear Authors,

Thank you for your thoughtful responses to the comments I provided on the revised manuscript.

After careful consideration, I am satisfied with the revisions you have made and your responses to my concerns. Therefore, I am confirming my acceptance of the manuscript for publication. I believe your efforts have improved the quality of the paper, and I look forward to seeing it contribute to the scientific community.

Best regards,

Reviewer

7. PLOS authors have the option to publish the peer review history of their article (what does this mean? ). If published, this will include your full peer review and any attached files.

**Do you want your identity to be public for this peer review?** For information about this choice, including consent withdrawal, please see our Privacy Policy .

Reviewer #2: No

---

## [Editor Report · Acceptance letter]

PONE-D-24-35309R3

PLOS ONE

Dear Dr. Katapally,

I'm pleased to inform you that your manuscript has been deemed suitable for publication in PLOS ONE. Congratulations! Your manuscript is now being handed over to our production team.

Kind regards,

on behalf of

Dr. Mohammad Jamil Rababa

Academic Editor

PLOS ONE